# Computational Mirrors: Blind Inverse Light Transport by Deep Matrix Factorization

**Miika Aittala**
MIT
miika@csail.mit.edu

**Prafull Sharma**
MIT
prafull@mit.edu

**Lukas Murmann**
MIT
lmurmann@mit.edu

**Adam B. Yedidia**
MIT
adamy@mit.edu

**Gregory W. Wornell**
MIT
gww@mit.edu

**William T. Freeman**
MIT, Google Research
billf@mit.edu

**Frédo Durand**
MIT
fredo@mit.edu

## Abstract

We recover a video of the motion taking place in a hidden scene by observing changes in indirect illumination in a nearby uncalibrated visible region. We solve this problem by factoring the observed video into a matrix product between the unknown hidden scene video and an unknown light transport matrix. This task is extremely ill-posed as any non-negative factorization will satisfy the data. Inspired by recent work on the Deep Image Prior, we parameterize the factor matrices using randomly initialized convolutional neural networks trained in a one-off manner, and show that this results in decompositions that reflect the true motion in the hidden scene.

## 1   Introduction

We study the problem of recovering a video of activity taking place outside our field of view by observing its indirect effect on shadows and shading in an observed region. This allows us to turn, for example, the pile of clutter in Figure 1 into a "computational mirror" with a low-resolution view into non-visible parts of the room.

The physics of light transport tells us that the image observed on the clutter is related to the hidden image by a linear transformation. If we were to know this transformation, we could solve for the hidden video by matrix inversion (we demonstrate this baseline approach in Section 3). Unfortunately, obtaining the transport matrix by measurement requires an expensive pre-calibration step and access to the scene setup. We instead tackle the hard problem of estimating *both* the hidden video and the transport matrix simultaneously from a single input video of the visible scene. For this, we cast the problem as matrix factorization of the observed clutter video into a product of an unknown transport matrix and an unknown hidden video matrix.

Matrix factorization is known to be very ill-posed. Factorizations for any matrix are in principle easy to find: we can simply choose one of the factors at will (as a full-rank matrix) and recover a compatible factor by pseudoinversion. Unfortunately, the vast majority of these factorizations are meaningless for a particular problem. The general strategy for finding *meaningful* factors is to impose problem-dependent priors or constraints — for example, non-negativity or spatial smoothness. While successful in many applications, meaningful image priors can be hard to express computationally. In particular, we are not aware of any successful demonstrations of the inverse light transport factorization problem in the literature. We find that classical factorization approaches produce solutions that are scrambled beyond recognition.

Our key insight is to build on the recently developed Deep Image Prior [37] to generate the factor matrices as outputs of a pair of convolutional neural networks trained in a one-off fashion. That is, we randomly initialize the neural net weights and inputs and then perform a gradient descent to find a set of weights such that their outputs' matrix product yields the known video. No training data or pre-training is involved in the process. Rather, the structure of convolutional neural networks, alternating convolutions and nonlinear transformations, induces a bias towards factor matrices that exhibit consistent image-like structure, resulting in recovered videos that closely match the hidden scene, although global ambiguities such as reflections and rotations remain. We found that this holds true of the video factor as well as the transport factor, in which columns represent the scene's response to an impulse in the hidden scene, and exhibit image-like qualities.

The source code, supplemental material, and a video demonstrating the results can be found at the project webpage at `compmirrors.csail.mit.edu`.

## 2 Related Work

**Matrix Factorization.** Matrix factorization is a fundamental topic in computer science and mathematics. Indeed, many widely used matrix transformations and decompositions, such as the singular value decomposition, eigendecomposition, and LU decomposition, are instances of constrained matrix factorization. There has been extensive research in the field of blind or lightly constrained matrix factorization. The problem has applications in facial and object recognition [19], sound separation [40], representation learning [36], and automatic recommendations [44]. Neural nets have been used extensively in this field [11, 28, 29, 17], and are often for matrix completion with low-rank assumption [36, 44].

Blind deconvolution [20, 18, 5, 21, 31, 12] is closely related to our work but involves a more restricted class of matrices. This greatly reduces the number of unknowns (a kernel rather than a full matrix) and makes the problem less ill-posed, although still quite challenging.

Koenderink et al. [16] analyze a class of problems where one seeks to simultaneously estimate some property, and calibrate the linear relationship between that property and the available observations. Our blind inverse light transport problem falls into this framework.

**Deep Image Prior.** In 2018, Ulyanov et al. [37] published their work on the Deep Image Prior—the remarkable discovery that due to their structure, convolutional neural nets inherently impose a natural-image-like prior on the outputs they generate, even when they are initialized with random weights and without any pretraining. Since the publication of [37], there have been several other papers that make use of the Deep Image Prior for a variety of applications, including compressed sensing [38], image decomposition [6], denoising [4], and image compression [9]. In concurrent work, the Deep Image Prior and related ideas have also been applied to blind deconvolution [1, 27].

**Light Transport Measurement.** There has been extensive past work on measuring and approximating light transport matrices using a variety of techniques, including compressed sensing [25], kernel Nyström methods [41], and Monte Carlo methods [33]. [32] and [7] study the recovery of an image's reflectance field, which is the light transport matrix between the incident and exitant light fields.

**Non-Line-of-Sight (NLoS) Imaging.** Past work in *active* NLoS imaging focuses primarily on active techniques using time-of-flight information to resolve scenes [43, 34, 10]. Time-of-flight information allows the recovery of a wealth of information about hidden scenes, including number of people [42], object tracking [15], and general 3D structure [8, 24, 39]. In contrast, past work in *passive* NLoS imaging has focused primarily on occluder-based imaging methods. These imaging methods can simply treat objects in the surrounding environment as pinspecks or pinholes to reconstruct hidden scenes, as in [35] or [30]. Others have used corners to get 1D reconstructions of moving scenes [3], or used sophisticated models of occluders to infer light fields [2].

## 3 Inverse Light Transport

We preface the development of our factorization method by introducing the inverse light transport problem, and presenting numerical real-world experiments with a classical matrix inversion solution when the transport matrix is known. In later sections we study the case of *unknown* transport matrices.

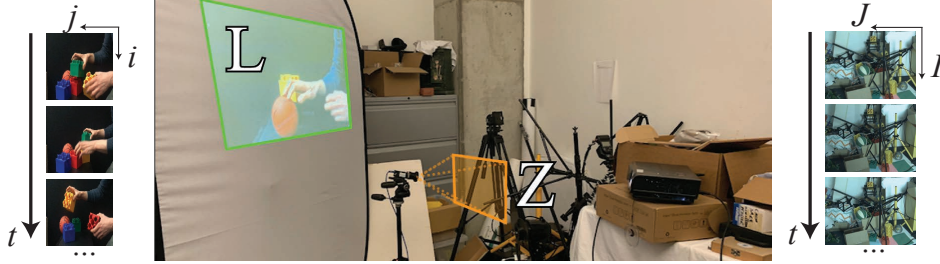

Figure 1: A typical experimental setup used throughout this paper. A camera views a pile of clutter, while a hidden video $L$ is being projected outside the direct view $Z$ of the camera. We wish to recover the hidden video from the shadows and shading observed on the clutter. The room lights in this photograph were turned on for the purpose of visualization only. During regular capture, we try to minimize any sources of ambient light. We encourage the reader to view the supplemental video to see the data and our results in motion.

## 3.1 Problem Formulation

The problem addressed in this paper is illustrated in Figure 1. We observe a video $Z$ of, for example, a pile of clutter, while a hidden video $L$ plays on a projector behind the camera. Our aim is to recover $L$ by observing the subtle changes in illumination it causes in the observed part of the scene. Such cues include e.g. shading variations, and the clutter casting moving shadows in ways that depend on the distribution of incoming light (i.e. the latent image of the surroundings). The problem statement discussed here holds in more general situations than just the projector-clutter setup, but we focus on it to simplify the exposition and experimentation.

The key property of light transport we use is that it is *linear* [13, 26]: if we light up two pixels on the hidden projector image in turn and sum the respective observed images of the clutter scene, the resulting image is the same as if we had taken a single photograph with both pixels lit at once. More generally, for every pixel in the hidden projector image, there is a corresponding response image in the observed scene. When an image is displayed on the projector, the observed image is a weighted sum of these responses, with weights given by the intensities of the pixels in the projector image. In other words, the image formation is the matrix product

$$Z = TL \tag{1}$$

where $Z \in \mathbb{R}^{IJ \times t}$ is the observed image of resolution $I * J$ at $t$ time instances, and $L \in \mathbb{R}^{ij \times t}$ is the hidden video of resolution $i * j$ (of same length $t$), and the light transport matrix $T \in \mathbb{R}^{IJ \times ij}$ contains all of the response images in its columns.[1] $T$ has no dependence on time, because the camera, scene geometry, and reflectance are static.

This equation is the model we work with in the remainder of the paper. In the subsequent sections, we make heavy use of the fact that all of these matrices exhibit image-like spatial (and temporal) coherence across their dimensions. A useful intuition for understanding the matrix $T$ is that by viewing each of its columns in turn, one would see the scene as if illuminated by just a single pixel of the hidden image.

## 3.2 Inversion with Known Light Transport Matrix

We first describe a baseline method for inferring the hidden video from the observed one in the non-blind case. In addition to the observed video $Z$, we assume that we have previously measured the light transport $T$ by lighting up the projector pixels individually in sequence and recording the response in the observed scene. We discretize the projector into $32 \times 32$ square pixels, corresponding to $i = j = 32$. As we now know two of the three matrices in Equation 1, we can recover $L$ as a solution to the least-squares problem $\mathrm{argmin}_L ||TL - Z||_2^2$. We augment this with a penalty on spatial

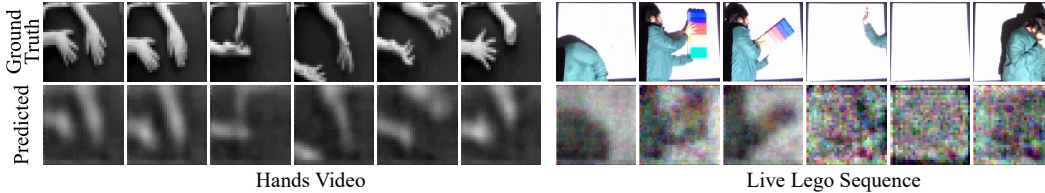

Figure 2: Reconstructions with known light transport matrix.

gradients. In practice, we measure $T$ in a mixed DCT and PCA basis to improve the signal-to-noise ratio, and invert the basis transformation post-solve. We also subtract a black frame with all projector pixels off from all measured images to eliminate the effect of ambient light.

Figure 2 shows the result of this inversion for two datasets, one where a video was projected on the wall, and the other in a live-action scenario. The recovered video matches the ground truth, although unsurprisingly it is less sharp. This non-blind solution provides a sense of upper bound for the much harder blind case.

The amount of detail recoverable depends on the content of the observed scene and the corresponding transport matrix [23]. In this and subsequent experiments, we consider the case where the objects in the observed scene have some mutual occlusion, so that they cast shadows onto one another. This improves the conditioning of the light transport matrix. In contrast, for example, a plain diffuse wall with no occluders would act as a strong low-pass filter and eliminate the high-frequency details in the hidden video. However, we stress that we do not rely on explicit knowledge of the geometry, nor on *direct* shadows being cast by the hidden scene onto the observed scene.

## 4 Deep Image Prior based Matrix Factorization

Our goal is to recover the latent factors when we *do not* know the light transport matrix. In this section, we describe a novel matrix factorization method that uses the Deep Image Prior [37] to encourage natural-image-like structure in the factor matrices. We first describe numerical experiments with one-dimensional toy versions of the light transport problem, as well as general image-like matrices. We also demonstrate the failure of classical methods to solve this problem. Applications to real light transport configurations will be described in the next section.

### 4.1 Problem Formulation

In many inference problems, it is known that the observed quantities are formed as a product of latent matrices, and the task is to recover these factors. Matrix factorization techniques seek to decompose a matrix $Z$ into a product $Z \approx TL$ (using our naming conventions from Section 3), either exactly or approximately. The key difficulty is that a very large space of different factorizations satisfy any given matrix.

The most common solution is to impose additional priors on the factors, for example, non-negativity [19, 40] and spatial continuity [21]. They are tailored on a per-problem basis, and often capture the desired properties (e.g. likeness to a natural image) only in a loose sense. Much of the work in nonnegative matrix factorization assumes that the factor matrices are low-rank rather than image-like.

The combination of being severely ill-posed and non-convex makes matrix factorization highly sensitive to not only the initial guess, but also the *dynamics* of the optimization. While this makes analysis hard, it also presents an opportunity: by shaping the loss landscape via suitable parameterization, we can guide the steps towards better local optima. This is a key motivation behind our method.

### 4.2 Method

We are inspired by the Deep Image Prior [37] and Double-DIP [6], where an image or a pair of images is parameterized via convolutional neural networks that are optimized in a one-off manner for each test instance. We propose to use a pair of one-off trained CNNs to generate the two factor matrices

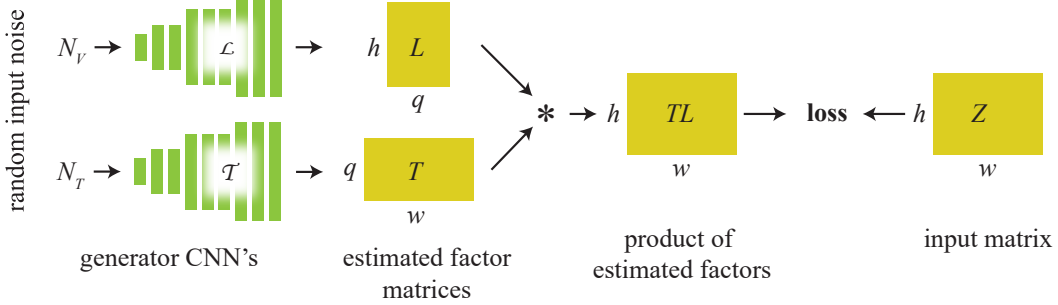

Figure 3: High level overview of our matrix factorization approach. The CNNs are initialized randomly and "overfitted" to map two vectors of noise onto two matrices $T$ and $L$, with the goal of making their product match the input matrix $Z$. In contrast to optimizing directly for the entries of $T$ and $L$, this procedure regularizes the factorization to prefer image-like structure in these matrices.

in our problem (Figure 3). We start with two randomly initialized CNNs, each one outputting a respective matrix $L$ and $T$. Similarly to [37], these CNNs are not trained from pairs of input/output labeled data, but are trained only once and specifically to the one target matrix. The optimization adjusts the weights of these networks with the objective of making the product of their output matrices identical to the target matrix being factorized. The key idea is that the composition of convolutions and pointwise nonlinearities has an inductive bias towards generating image-like structure, and therefore is more likely to result in factors that have the appearance of natural images. The general formulation of our method is the minimization problem

$$\mathrm{argmin}_{\theta,\phi} d(\mathcal{T}(N_T;\theta)\mathcal{L}(N_L;\phi), Z) \qquad (2)$$

where $Z \in \mathbb{R}^{h \times w}$ is the matrix we are looking to factorize, and $\mathcal{T} : \mathbb{R}^{n_T} \mapsto \mathbb{R}^{h \times q}$ and $\mathcal{L} : \mathbb{R}^{n_L} \mapsto \mathbb{R}^{q \times w}$ are functions implemented by convolutional neural networks, parametrized by weights $\theta$ and $\phi$, respectively. These are the optimization variables. $q$ is a chosen inner dimension of the factors (by default the full rank choice $q = \min(w, h)$). $d : \mathbb{R}^{w \times h} \times \mathbb{R}^{w \times h} \mapsto \mathbb{R}$ is any problem-specific loss function, e.g. a pointwise difference between matrices. The inputs $N_T \in \mathbb{R}^{n_T}$ and $N_L \in \mathbb{R}^{n_L}$ to the networks are typically fixed vectors of random noise. Optionally the values of these vectors may be set as learnable parameters. They can also subsume other problem-specific inputs to the network, e.g. when one has access to auxiliary images that may guide the network in performing its task. The exact design of the networks and the loss function is problem-specific.

### 4.3 Experiments and Results

We test the CNN-based factorization approach on synthetically generated tasks, where the input is a product of a pair of known ground truth matrices. We use both toy data that simulates the characteristics of light transport and video matrices, as well as general natural images.

We design the generator neural networks $\mathcal{T}$ and $\mathcal{L}$ as identically structured sequences of convolutions, nonlinearities and upsampling layers, detailed in the supplemental appendix. To ensure non-negativity, the final layer activations are exponential functions. Inspired by [22], we found it useful to inject the pixel coordinates as auxiliary feature channels on every layer. Our loss function is $d(x, y) = ||\nabla(x - y)||_1 + w||x - y||_1$ where $\nabla$ is the finite difference operator along the spatial dimensions, and $w$ is a small weight; the heavy emphasis on local image gradients appears to aid the optimization. We use Adam [14] as the optimization algorithm. The details can be found in the supplemental appendix.

Figure 4 shows a pair of factorization results, demonstrating that our method is able to extract images similar to the original factors. We are not aware of similar results in the literature; as a baseline we attempt these factorizations with the DIP disabled, and with standard non-negative matrix factorization. These methods fail to produce meaningful results.

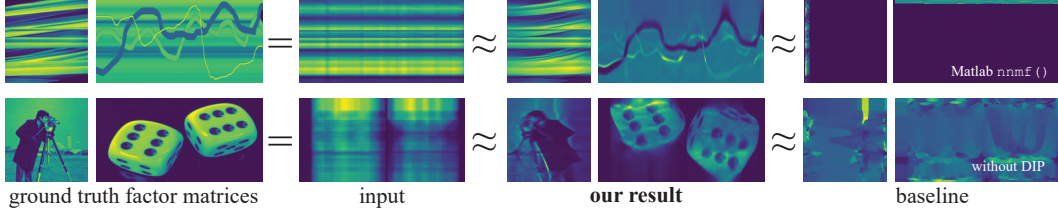

ground truth factor matrices          input          **our result**          baseline

Figure 4: Matrix factorization results. The input to the method is a product of the two leftmost matrices. Our method finds visually readable factors, and e.g. recovers all three of the faint curves on the first example. On the right, we show two different baselines: one computed with Matlab's non-negative matrix factorization (in alternating least squares mode), and one using our code but optimizing directly on matrix entries instead of using the CNN, with an $L_1$ smoothness prior.

## 4.4 Distortions and Failure Modes

The factor matrices are often warped or flipped. This stems from ambiguities in the factorization task, as the factor matrices can express mutually cancelling distortions. However, the DIP tends to strongly discourage distortions that break spatial continuity and scramble the images.

More specifically, the space of ambiguities and possible distortions can be characterized as follows [16]. Let $T_0$ and $L_0$ be the true underlying factors, the observed video thus being $Z = T_0 L_0$. All valid factorizations are then of the form $T = T_0 A^\dagger$ and $L = A L_0$, where $A$ is an arbitrary full-rank matrix, and $A^\dagger$ is its inverse. This can be seen by substituting $TL = (T_0 A^\dagger)(A L_0) = T_0(A^\dagger A)L_0 = T_0 L_0 = Z$.

The result of any factorization implicitly corresponds to some choice of $A$ (and $A^\dagger$). In simple and relatively harmless cases (in that they do not destroy the legibility of the images), the matrix $A$ can represent e.g. a permutation that flips the image, whence $A^\dagger$ is a flip that restores the original orientation. They can also represent reciprocal intensity modulations, meaning that there is a fundamental ambiguity about the intensity of the factors. However, for classical factorization methods, the matrices tend to consist of unstructured "noise" that scrambles the image-like structure in $T_0$ and $L_0$ beyond recognition. Our finding is that the use of DIP discourages such factorizations.

## 5 Blind Light Transport Factorization

We now combine the ideas from the previous two sections and present a method for tackling the inverse light transport problem blindly, when we have no access to a measured light transport matrix. We show results on both synthetic and real data, and study the behavior of the method experimentally.

### 5.1 Method

**Setup**  Continuing from Section 3, our goal is to factor the observed video $Z \in \mathbb{R}^{IJ \times t}$ of $I * J$ pixels and $t$ frames, into a product of two matrices: the light transport $T \in \mathbb{R}^{IJ \times ij}$, and the hidden video $L \in \mathbb{R}^{ij \times t}$. The hidden video is of resolution $i * j$, with $i = j = 16$. Most of our input videos are of size $I = 96$ (height), $J = 128$ (width), and $t$ ranging from roughly 500 to 1500 frames.

Following our approach in Section 4, the task calls for designing two convolutional neural networks that generate the respective matrices. Note that $T$ can be viewed as a 4-dimensional $I \times J \times i \times j$ tensor, and likewise $L$ can be seen as a 3-dimensional $i \times j \times t$ tensor. We design the CNNs to generate the tensors in these shapes, and in a subsequent network operation reshape the results into the stacked matrix representation, so as to evaluate the matrix product. The dimensionality of the convolutional filters determines which dimensions in the result are bound together with image structure. In the following, we describe the networks generating the factors. An overview of our architecture is shown in Figure 5.

**Hidden Video Generator Network**  The hidden video tensor $L$ should exhibit image-like structure along all of its three dimensions. Therefore a natural model is to use 3D convolutional kernels in the network $\mathcal{L}$ that generates it. Aside from its dimensionality, the network follows a similar sequential

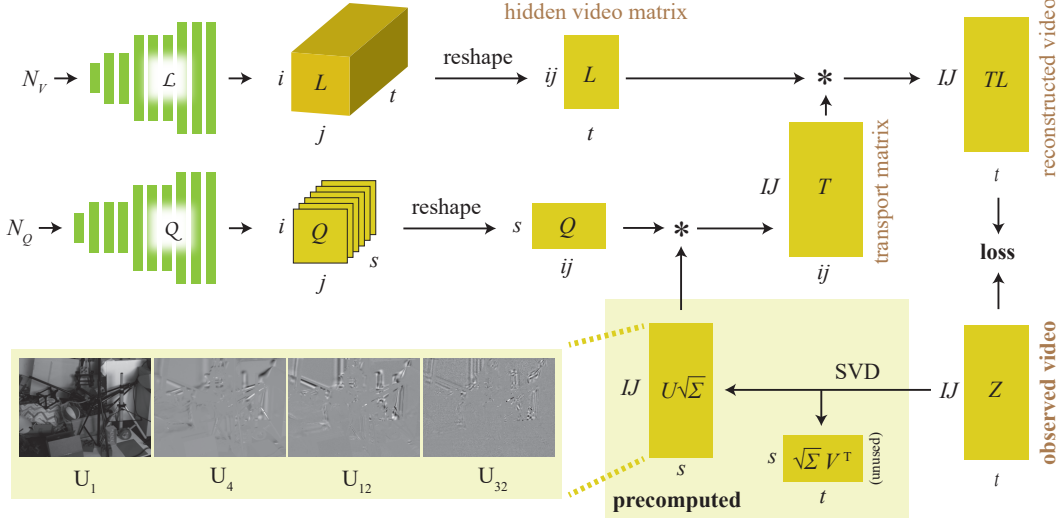

Figure 5: An overview of the architecture and data flow of our blind inverse light transport method. Also shown (bottom left) are examples of the left singular vectors stored in $U$. $\mathcal{L}$ and $\mathcal{Q}$ are convolutional neural networks, and the remainder of the blocks are either multidimensional tensors or matrices, with dimensions shown at the edges. The matrices in the shaded region are computed once during initialization. The input $Z$ to the method is shown in the lower right corner.

up-scaling design as that discussed in Section 4. It is illustrated in Figure 5 and detailed in the supplemental appendix.

**Light Transport Generator Network** The light transport tensor $T$, likewise, exhibits image structure between all its dimensions, which in principle would call for use of 4D convolutions. Unfortunately these are very slow to evaluate, and unimplemented in most CNN frameworks. We also initially experimented with alternating between 2D convolutions along $I, J$ dimensions and $i, j$ dimensions, and otherwise following the same sequential up-scaling design. While we reached some success with this design, we found a markedly different architecture to work better.

The idea is to express the slices of $T$ as linear combinations of basis images obtained from the singular value decomposition (SVD) of the input video. This is both computationally efficient and guides the optimization by constraining the iterates and the solution to lie in the subspace of valid factorizations. Intuitively, the basis expresses a frequency-like decomposition of shadow motions and other effects in the video, as shown in Figure 5.

We begin by precomputing the truncated singular value decomposition $U\Sigma V^T$ of the input video $Z$ (with the highest $s = 32$ singular values), and aim to express the columns of $T$ as linear combinations of the left singular vectors $U \in \mathbb{R}^{IJ \times s}$. The individual singular vectors have the dimensions $I \times J$ of the input video. These vectors form an appropriate basis for constructing the physical impulse response images in $T$, as the the column space of $Z$ coincides with that of $T$ due to them being related by right-multiplication. [2]

We denote the linear combination by a matrix $Q \in \mathbb{R}^{s \times ij}$. The task boils down to finding $Q$ such that $(UQ)L \approx Z$. Here $L$ comes from the DIP-CNN described earlier. While one could optimize for the entries of $Q$ directly, we again found that generating $Q$ using a CNN produced significantly improved results. For this purpose, we use a CNN that performs 2D convolutions in the $ij$-dimension, but *not* across $s$, as only the former dimension is image-valued. In summary, the full minimization problem becomes a variant of Eq. 2:

$$\mathrm{argmin}_{\theta,\phi} d(U\sqrt{\Sigma}\mathcal{Q}(N_Q;\theta)\mathcal{L}(N_L;\phi), Z) \tag{3}$$

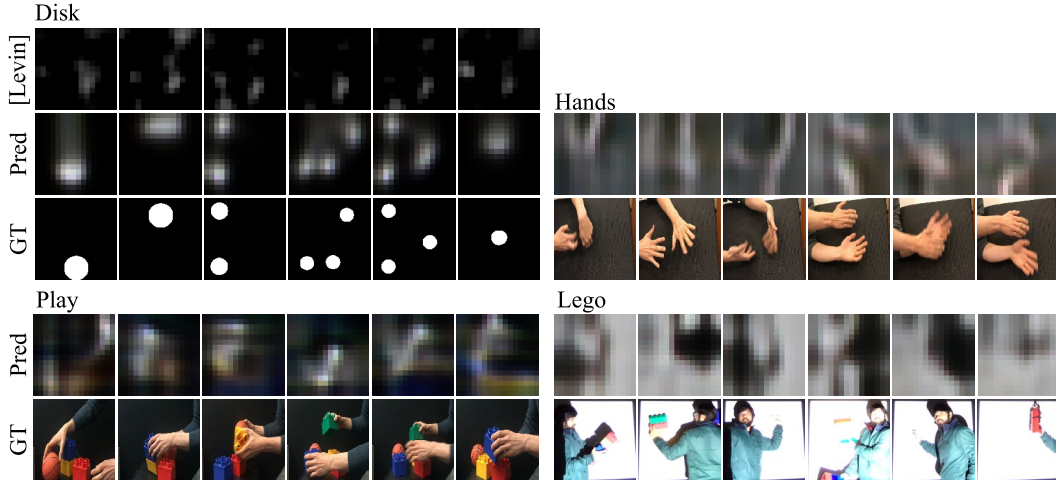

Figure 6: Blind light transport factorization using our method. The first three sequences are projected onto a wall behind the camera. The *Lego* sequence is performed live in front of the illuminated wall.

where $\mathcal{Q}$ implements the said CNN. The somewhat inconsequential additional scaling term $\sqrt{\Sigma}$ originates from our choice to distribute the singular value magnitudes equally between the left and right singular vectors.

**Implementation Details** The optimization is run using Adam [14] algorithm, simultaneously optimizing over parameters of $\mathcal{Q}$ and $\mathcal{L}$. The loss function is a sum of pointwise difference and a heavily weighted temporal gradient difference between $Z$ and reconstructed $TL$. Details are in the supplemental appendix. We extend the method to color by effectively treating it as three separate problems for R, G, and B; however, they become closely tied as the channels are generated by the same neural network as 3-dimensional output. We also penalize color saturation in the transport matrix to encourage the network to explain colors with the hidden video. To ensure non-negativity, we use a combination of exponentiations and tanh functions as output activations for the network $\mathcal{L}$. For $T$ we penalize negative values with a prior. We also found it useful to inject pixel and time coordinates as auxiliary feature maps, and to multiply a Hann window function onto intermediate feature maps at certain intermediate layers. These introduce forced spatial variation into the feature maps and help the networks to rapidly break symmetry in early iterations.

## 5.2 Experiments and Results

We test our method with multiple video datasets collected using a projector setup (as described in Section 3) recorded in different scenes with different hidden projected videos (Figure 6). We encourage the reader to view the supplemental video, as motion is the main focus of this work.

The results demonstrate that our method is capable of disentangling the light transport from the content of the hidden video to produce a readable estimate of the latter. The *disk* dataset is a controlled video showing variously complex motions of multiple bright spots. The number of the disks and their related positions are resolved correctly, up to a spatial warp ambiguity similar to the one discussed in Section 4.4. The space of ambiguities in this full 2-dimensional scenario is significantly larger than in the 1-D factorization: the videos can be arbitrarily rotated, flipped, shifted and often exhibit some degree of non-linear warping. The color balance between the factors is also ambiguous. As a control for possible unforeseen nonlinearities in the experimental imaging pipeline, we also tested the method on a semi-synthetic dataset that was generated by explicitly multiplying a measured light transport matrix with the *disk* video; the results from this synthetic experiment were essentially identical to our experimental results.

The other hidden videos in our test set exhibit various degrees of complexity. For example, in *hands*, we wave a pair of hands back and forth; watching our solved video, the motions and hand gestures are clearly recognizable. However, as the scenes become more complex, such as in a long fast-forwarded video showing colored blocks being variously manipulated (*play*), the recovered video does show

similarly colored elements moving in correlation with the ground truth, but the overall action is less intelligible.

We also test our method on the live-action sequence introduced in Section 3. Note that in this scenario the projector plays no role, other than acting as a lamp illuminating the scene. While less clear than the baseline solution with a measured transport matrix, our blindly factored solution does still resolve the large-scale movements of the person, including movements of limbs as he waves his hands and rotates the Lego blocks.

**Comparison with Existing Approaches**    We compare our method to an extension of the deblurring approach by Levin et al. in [21]. We believe that blind deconvolution is the closest problem to ours, since it can be seen as a matrix factorization between a convolution matrix and a latent sharp image. We extended their marginalization method to handle general matrices and not just convolution, and use the same sparse derivative prior as them (see the supplementary materials for more details on how we adapted the approach). Figure 6 and the supplementary video show that this approach produces vastly inferior reconstructions.

# 6   Discussion and Conclusions

We have shown that cluttered scenes can be computationally turned into low-resolution mirrors without prior calibration. Given a single input video of the visible scene, we can recover a latent video of the hidden scene as well as a light transport matrix. We have expressed the problem as a factorization of the input video into a transport matrix and a lighting video, and used a deep prior consisting of convolutional neural networks trained in a one-off fashion. We find it remarkable that merely asking for latent factors easily expressible by a CNN is sufficient to solve our problem, allowing us to entirely bypass challenges such as the estimation of the geometry and reflectance properties of the scene.

Blind inverse light transport is an instance of a more general pattern, where the latent variables of interest (the video) are tangled with another set of latent variables (the light transport), and to get one, we must simultaneously estimate both [16]. Our approach shows that when applicable, identifying and enforcing natural image structure in both terms is a powerful tool. We hope that our method can inspire novel approaches to a wide range of other apparently hopelessly ill-posed problems.

**Acknowledgements**

This work was supported, in part, by DARPA under Contract No. HR0011-16-C-0030, and by NSF under Grant No. CCF-1816209. The authors wish to thank Luke Anderson for proofreading and helping with the manuscript.

## Footnotes

[1]Throughout this paper, we use notation such as $T \in \mathbb{R}^{IJ \times ij}$ to imply that $T$ is, in principle, a 4-dimensional tensor with dimensions $I$, $J$, $i$ and $j$, and that we have packed it into a 2-dimensional tensor (matrix) by stacking the $I$ and $J$ dimensions together into the columns, and $i$ and $j$ in the rows. Thus, when we refer to e.g. columns of this matrix as images, we are really referring to the unpacked $I \times J$ array corresponding to that column.

[2]Strictly speaking, some dimensions of the true $T$ may be lost in the numerical null space of $Z$ (or to the truncated singular vectors) if the light transport "blurs" the image sufficiently, making it impossible to exactly reproduce the $T$ from $U$. In practice we find that at the resolutions we realistically target, this does not prevent us from obtaining meaningful factorizations.

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
