[Supplementary Material · compmirrors_supplementary.pdf]

# Supplemental Appendix to:
# Computational Mirrors: Blind Inverse Light Transport by Deep Matrix Factorization

## 1   Implementation Details

In the following, we discuss the technical implementation of the networks and losses presented in the paper. Please refer to the associated code release for the exact implementation details.

### 1.1   Network Architectures

The detailed architecture of the three networks used is discussed in Table 1 (the 1D matrix factorization task), Table 2 (the video prediction network $\mathcal{L}$) and Table 3 (the light transport singular vector weight prediction network $\mathcal{Q}$). Each network consists of a linear chain of layers, alternating between convolutions and upsampling. In particular, there are no downsampling cycles or skip connections.

### 1.2   Loss and Training Details for Section 5.1

The data fit loss the full blind inverse light transport problem is a sum of the following:

- Direct data fit loss $0.01\,||TL - Z||_2^2$
- Temporal gradient fit loss $||\nabla_t(TL - Z)||_1$ where $\nabla_t$ evaluates the (unnormalized) finite difference across dimension $t$. The finite difference interval is chosen at random on each iteration in $[1, 8]$, so as to consider multiple time scales.

Priors are added to this loss:

- Nonnegativity prior $10\,||\min(T, 0)||_2$ over the transport matrix entries.
- Smoothness prior $0.001\,||\nabla_{IJ}T||_1$ over the observed image dimensions in the transport matrix $T$.
- Color saturation prior $0.001||T - \mathrm{mean}_c(T)||_1$ that penalizes the difference of the transport matrix RGB channels from their average.
- Magnitude prior $0.0001\,||Q_0||_1$ where $Q_0$ are the weights for the first singular vector, i.e. the residual amount added on top of the mean. This prior merely anchors the magnitudes of the solution loosely, as $T$ and $L$ could accumulate arbitrary reciprocal multipliers over the iterations.

During the data loading phase, we identify any pixels that are overexposed (saturated) in any of the input frames, and exclude these pixels from all subsequent computations.

The colors are handled by predicting three output channels (R, G and B) in each network, and performing the matrix multiplication separately in each color channel. The basis singular vectors $U$ are also evaluated separately for each color channel. While in theory this would allow for mutually unrelated solutions for the three channels, in practice the color features naturally fall into coinciding positions, as they are built from common internal network features.

We train both networks simultaneously using the Adam optimizer [1] with learning rate 0.00006.

We typically run the network for 100 000 iterations, which takes approximately four hours on an NVIDIA Titan Xp GPU. Typically we see coarse results at a few thousand iterations, and details become filled in over the remaining iterations. We implemented our model using PyTorch [3].

## 2 Comparison to Levin et al. [2]

In Section 5.2 we compared the results achieved by our method to that achieved by an adapted version of the approach used by Levin et al. in [2]. We believe that this represents the closest existing analogue to a solution to our problem, since blind deconvolution can be seen as equivalent to matrix factoring when the light transport matrix must be Toeplitz, and in the problem of [2] both the base image and the blur kernel must exhibit spatial smoothness properties.

Following the example of [2], we used an Expectation-Maximization (EM) framework to perform the joint reconstruction of the light transport matrix and the scene movie. In the E-step we solve for the mean light transport matrix given the scene movie, alone with an estimate of the associated covariance matrix over entries of the light transport matrix (for computational efficiency's sake, we assumed this covariance matrix was diagonal). In the M-step we solve for the best scene movie given the distribution over the light transport matrices implied by the mean and variance recovered in the E-step. In both steps, we penalized high spatial variations. We randomly initialized the light transport matrix and the scene movie, and iterated until the result converged.

| id | type | output features | filter size | activation |
|---|---|---|---|---|
| 1 | conv | 32 | $4 \times 4$ | tanh |
| 2 | upsample | | | |
| 3 | conv | 64 | $4 \times 4$ | tanh |
| 4 | upsample | | | |
| 5 | conv | 64 | $4 \times 4$ | tanh |
| 6 | upsample | | | |
| 7 | conv | 128 | $4 \times 4$ | tanh |
| 8 | upsample | | | |
| 9 | conv | 128 | $4 \times 4$ | tanh |
| 10 | upsample | | | |
| 11 | conv | 64 | $3 \times 3$ | leaky relu |
| 12 | conv | 1 | $3 \times 3$ | exp |

Table 1: Network architecture for $\mathcal{T}$ and $\mathcal{L}$ in Section 4.3. (plain matrix factorization, or "1D light transport"). Both networks are essentially identical in architecture. The input to the network is a random 64-channel normal distributed tensor with $1/32$'th the spatial dimension of the target matrix. We include it as a learnable parameter of the optimization. The upsampling layers use bilinear interpolation. On every layer, we concatenate a pair of horizontal and vertical linear gradients (representing x and y coordinate values) as input feature maps. In the second-to-last layer, we concatenate as an auxiliary feature map the average of the input matrix rows. This corresponds to the average frame of the input "video", when that is the interpretation of the data. We also apply a pixel-wise dropout of probability $0.5$ on the input noise features.

| id | type | features | filter size | activation |
|---|---|---|---|---|
| 1 | conv | 64 | $3 \times 3 \times 3$ | leaky relu |
| 2 | upsample | | | |
| 3 | conv | 64 | $3 \times 3 \times 3$ | leaky relu |
| 4 | conv | 64 | $3 \times 3 \times 3$ | leaky relu |
| 5 | upsample | | | |
| 6 | conv | 64 | $3 \times 3 \times 3$ | leaky relu |
| 7 | conv | 64 | $3 \times 3 \times 3$ | leaky relu |
| 8 | conv | 64 | $3 \times 3 \times 3$ | leaky relu |
| 9 | upsample | | | |
| 10 | conv | 64 | $3 \times 3 \times 3$ | leaky relu |
| 11 | conv | 32 | $3 \times 3 \times 3$ | leaky relu |
| 12 | conv | 6 | $3 \times 3 \times 3$ | exp + tanh |

Table 2: Network architecture for the hidden video network $\mathcal{L}$ in Section 5.1. The input to the network is a learnable random normal initialized tensor of 4 feature channels and $t/8 \times 2 \times 2$ (i.e. $2 \times 2$ pixels times $1/8$'th of the length of the video). The upsampling layers use nearest neighbor interpolation. Up to layer 7, all convolutional layers append three coordinate feature maps across space and time (with range $[-1, 1]$). The final layer outputs two RGB images – one is passed to an exponential function and the other to a $b(\frac{1}{2} + \frac{1}{2}\tanh(x))$, where $b$ is a learnable RGB "blacklevel" parameter of dimension 3. The output of the tanh and exp are summed into the final predicted tensor $L$. The goal of the final activations is to prevent negative outputs by construction, and to bias the network towards using the learned black level as the "default background color" onto which it adds and subtracts image content. The output of the network is scaled by the reciprocal of the number of pixels $1/(16 * 16)$. The leaky relu slope parameter is $0.1$.

| id | type | features | filter size | activation |
|---|---|---|---|---|
| 1 | conv | 32 | $3 \times 3$ | leaky relu |
| 2 | upsample | | | |
| 4 | conv | 64 | $3 \times 3$ | leaky relu |
| 5 | conv | 64 | $3 \times 3$ | leaky relu |
| 6 | conv | 64 | $3 \times 3$ | leaky relu |
| 7 | upsample | | | |
| 8 | conv | 64 | $3 \times 3$ | leaky relu |
| 9 | conv | 64 | $3 \times 3$ | leaky relu |
| 10 | conv | 64 | $3 \times 3$ | leaky relu |
| 11 | upsample | | | |
| 12 | conv | 128 | $3 \times 3$ | leaky relu |
| 13 | conv | 256 | $3 \times 3$ | leaky relu |
| 14 | conv | 96 | $3 \times 3$ | linear |

Table 3: Network architecture for the singular value weight network $\mathcal{Q}$ used to build the light transport matrix in Section 5.1. The input to the network is a learnable randomly initialized tensor of 32 feature channels at spatial dimensions $2 \times 2$ pixels. The upsampling layers use nearest neighbor interpolation. Up to layer 10, all convolutional layers append coordinate feature maps across the two spatial dimensions. In layers 8, 9 and 10 we also multiply a Hann window into the network outputs. The final layer outputs 32 RGB images packed into 96 channels, one corresponding to each singular vector. After the final layer, these channels are multiplied by 96 learnable weights intended to provide the optimizer with direct means to control their strength. These are initialized as square roots of the singular values. After the multiplication by the singular vector basis (see Section 5.1), a precomputed mean image of the input video is added onto every slice of $T$, so that the network only needs to learn the residual.