[Reviews · NeurIPS 2019]

Reviewer 1



I actually don't have much to say about this paper. It is clearly written and easy to understand. It presents compelling results which outperform existing techniques. I'm not sure why the title of the paper refers to "computational mirrors"---this feels out of place and it's not referred to in the paper except in the very last section (the conclusion). Also, since motion is the focus of the work, perhaps it would be worth mentioning "video" in the title? Perhaps something like "Blind Inverse Video Light Transport by Deep Matrix Factorization"?

Reviewer 2



Non-line-of-Sight imaging is recently receiving attention in the computer vision/computational photography community, for example, a paper regarding this received the CVPR2019 best paper award. Most existing researches on NLoS imaging are active. On the contrary, this paper is purely passive. It tries to factorize the observed video into a matrix product between unknown hidden scene video and an unknown light transport matrix. So, the problem setting of this paper is extremely challenging but interesting. Considering that the solution space is extremely huge, and popular priors of nonnegative constraints and spatial smoothness are shown to be insufficient. The authors proposed to use deep image prior, which seems to confine the solution space properly. Experiment results show that the factorization is quite nice in capturing motions in the illumination, yet less capable of recovering color and object details. I think this is reasonable, since the cluttered scene has obvious cascade shadows.

Reviewer 3



In this paper, the authors study the problem of reconstructing a hidden scene from the observed videos. The proposed method seeks to invert tight transport matrix without a calibration step. The problem is challenging and ill posed. The author learn a low-dimensional basis from observed videos and use deep image prior models for generating hidden scene and coefficients of the light transport basis. Originality: + The paper uses inverse light transport to recover a video of hidden scene without any calibration, which seems novel. + The idea of using deep image prior to model the light transport coefficients and hidden scene is also interesting. - The paper is missing comparison with other related techniques for reconstructing hidden scenes. Quality: + The paper is well written. + The paper provides experiments on real data. - The results presented in the paper are very weak. Even though the authors claim that they can see the motion of hidden objects, that seems convincing only in the example of moving discs. I am not convinced that the true scene or motions is reconstructed in the case of more complex examples. - The authors did not quantify the accuracy of their reconstruction in any meaningful way. Since the deep image prior or deep decoders tend to produce "realistic" images, the blurry patterns produced by them (that look kind of like images) alone can not be used as a sufficient evidence that the method is reconstructing the hidden scene. Clarity: + The paper is well-written and structured. The problem is clearly stated and formulated. Supplementary materials help to understand the experimental setup and significance of the problem. + I like the fact that the authors provided experimental results for both cases where the light transport matrix is known and the blind factorization where both T and L are unknown Significance: - The results are not significant or convincing in my opinion. - The reconstructed videos are too blurry to perform any reasonable computer vision or machine learning task. + I think the paper has some merits, but it is not complete. It could be a really strong paper if the authors had performed some analysis about the accuracy of the reconstruction or demonstrated that the recovered video contains some salient information about the hidden scene.

[Author Response · NeurIPS 2019]

We thank the reviewers for their time and their reviews. We address the questions below.

**Ambiguities beyond flips and rotations (R3)** As pointed out by R3, the albedo of the hidden scene is fundamentally ambiguous, as any intensity can be compensated by a reciprocal intensity in the transport matrix. To anchor the solution colors, we use the common "gray world assumption", and impose it by a simple chromaticity prior that discourages large differences between color channels. The color of the observed scene can therefore tint the colors of the hidden scene solution.

The space of ambiguities and potential distortions can be characterized as follows (see Koenderink et al., The Generic Bilinear Calibration-Estimation Problem). Let $T_0$ and $L_0$ be the true underlying factors, the observed video thus being $Z = T_0 L_0$. All "valid" factorizations are of the form $T = T_0 A^\dagger$ and $L = A L_0$, where A is chosen (almost) arbitrarily and $A^\dagger$ is its (pseudo)inverse. This can be seen by substituting $TL = (T_0 A^\dagger)(A L_0) = T_0 (A^\dagger A) L_0 = T_0 L_0 = Z$.

The result of any factorization implicitly corresponds to some choice of $A$ and $A^\dagger$. In simple cases, the matrix $A$ can represent e.g. a permutation that flips the image, whence $A^\dagger$ is a flip that restores the original orientation: this case is illustrated in Figure 4's conversely flipped matrices. They can also represent complementary color transformations as discussed above. However, for classical factorization methods, they tend to consist of unstructured "noise" that scrambles the image-like structure in $T_0$ and $L_0$ beyond recognition.

Our finding in the paper is that via DIP-based factorization, these transformations instead tend to express continuous and bijective image warps (and color modulations) that preserve the general image structure. As observed by R3, this does in practice include more complex distortions than just flips and rotations —- see for example the nonlinear stretching of the cameraman image in Figure 4. In full two dimensions, there is room for more complex distortions, but we still find that e.g. the relative motions of independent objects often remain readable.

**Geometric complexity (R3)** We assume that the scene contains a sufficient amount of geometric complexity to generate high-frequency features like shadows. This improves the conditioning of the problem, as discussed in the literature on frequency analysis of light transport effects (see e.g. A Theory of Locally Linear Light Transport by Mahajan et al.). We will emphasize this in the revised paper.

**Comparisons to previous methods (R4)** To our knowledge, no existing work attempts to solve the problem under a similarly general setup, with no assumptions about the shapes viewed in the scene. Attempts to use standard factorization methods consistently produce unstructured and scrambled results, analogous to the baselines in Figure 4. An example is seen in the supplemental video, where an SVD factorization is visualized at time 2:00 - 2:09.

To provide a comparison, we generalized a recent algorithm that addresses the closest analogue we could think of, i.e. blind deconvolution with a classical sparse gradient prior that models natural image statistics. As discussed in Section 5.2, we were unable to obtain competitive results despite fair efforts put into the experiment.

Regarding comparisons to other non-line-of-sight methods: active non-line-of-sight methods are outside the scope of the paper, as these techniques assume fundamentally different imaging modalities (usually static hidden scenes, actively probed over an extended period of time). Similarly, a recent passive non-line-of-sight technique by Bouman et al. (Turning Corners into Cameras: Principles and Methods, ICCV 2017) assumes a specific scene geometry with clearly defined "corners" and focuses on near-invisible signals, while our method assumes the geometry and reflectances of the relay objects are unknown.

**Validity of reconstructions for machine vision tasks (R4)** When factorizing light transport using traditional factorization techniques, there is no guarantee that the two factors correspond to the true visible and hidden scene, or even to any plausible image signal. The question raised by R4 is whether the factors reconstructed using DIP actually correspond to the true visible and hidden scenes, or are just arbitrary natural-looking images. Figure 6 of the paper provides a partial answer to that question. We show that for controlled scenes such as the Disks sequence, the reconstructed signal is clearly not arbitrary, but closely matches the ground truth sequence. Even for more complicated sequences (Hands), this correspondence still appears to hold.

[Meta-Review · NeurIPS 2019]

This paper has tackled an extremely challenging problem. It provides a neat and bold idea towards solving it. While the work is far from complete, as agreed upon by many of the reviewers in a discussion, it provides a first-cut idea and attempt, and enough detail to potentially carry this proof-of-concept further in the future. I suggest the authors carefully address the reviewer's comments.